# Unveiling a New Selenocyanate as a Multitarget Candidate with Anticancer, Antileishmanial and Antibacterial Potential

**DOI:** 10.3390/molecules27217477

**Published:** 2022-11-02

**Authors:** Sandra Ramos-Inza, Andreina Henriquez-Figuereo, Esther Moreno, Melibea Berzosa, Ignacio Encío, Daniel Plano, Carmen Sanmartín

**Affiliations:** 1Department of Pharmaceutical Technology and Chemistry, University of Navarra, Irunlarrea 1, E-31008 Pamplona, Spain; 2Instituto de Investigación Sanitaria de Navarra (IdiSNA), Irunlarrea 3, E-31008 Pamplona, Spain; 3Institute of Tropical Health of the University of Navarra (ISTUN), University of Navarra, Irunlarrea 1, E-31008 Pamplona, Spain; 4Department of Microbiology and Parasitology, University of Navarra, Irunlarrea 1, E-31008 Pamplona, Spain; 5Department of Health Sciences, Public University of Navarra, Avda. Barañain s/n, E-31008 Pamplona, Spain

**Keywords:** selenium, cytotoxicity, antileishmanial activity, antibacterial activity, antioxidant

## Abstract

Currently, cancer, leishmaniasis and bacterial infections represent a serious public health burden worldwide. Six cinnamyl and benzodioxyl derivatives incorporating selenium (Se) as selenocyanate, diselenide, or selenide were designed and synthesized through a nucleophilic substitution and/or a reduction using hydrides. Ferrocene was also incorporated by a Friedel–Crafts acylation. All the compounds were screened in vitro for their antiproliferative, antileishmanial, and antibacterial properties. Their capacity to scavenge free radicals was also assessed as a first approach to test their antioxidant activity. Benzodioxyl derivatives **2a**–**b** showed cytotoxicity against colon (HT-29) and lung (H1299) cancer cell lines, with IC_50_ values below 12 µM, and were also fairly selective when tested in nonmalignant cells. Selenocyanate compounds **1**–**2a** displayed potent antileishmanial activity in *L. major* and *L. infantum*, with IC_50_ values below 5 µM. They also exhibited antibacterial activity in six bacterial strains, notably in *S. epidermidis* with MIC and MBC values of 12.5 µg/mL. Ferrocene-containing selenide **2c** was also identified as a potent antileishmanial agent with radical scavenging activity. Remarkably, derivative **2a** with a selenocyanate moiety was found to act as a multitarget compound with antiproliferative, leishmanicidal, and antibacterial activities. Thus, the current work showed that **2a** could be an appealing scaffold to design potential therapeutic drugs for multiple pathologies.

## 1. Introduction

The cancer burden remains among the leading causes of death worldwide, mainly in high-income countries [1]. Concurrently, infectious diseases are major emerging threats to public health [2], causing serious issues in the successful prevention and treatment of diseases, especially in immunocompromised individuals such as cancer patients [3]. Pathogenic microorganisms can contribute to the development of the disease by excreting metabolites, activating inflammatory pathways, and stimulating an immune response [4]. Further, eleven species of pathogenic agents, including parasites and bacteria, have been known to be associated with certain types of cancer [5]. Likewise, the existence of common triggering factors reflects mutual features in the mechanisms underlying cancer and leishmaniasis [6]. Leishmaniasis is a neglected and poverty-associated disease caused by single-celled parasites of the genus *Leishmania* and transmitted to humans by phlebotomine female sandflies [7]. In this regard, DNA metabolism, protein kinase pathways, glucose catabolism enzymes, and polyamines metabolism could be some biochemical similarities shared by both parasites and cancer cells [8]. Current drug therapy for these diseases is limited by the high toxicity related to multiple side effects, and by the cancer cells [9], bacteria [10], and leishmaniasis [11] resistance to these clinically commercial drugs. Thus, the development of new, efficient therapeutic agents is urgently needed.

In this context, selenium (Se)-containing compounds have attracted great attention in the field of medicinal chemistry in recent years [12,13]. Se is an important element of biological molecules in bacteria and eukaryotes [14], and its key role in human health has been recognized [15]. Se is a pivotal component of selenoproteins, which are essential enzymes involved in a range of bioactivities related to the prevention and treatment of several diseases including cancer [16], mainly thought the protection against oxidative stress and maintenance of reactive oxygen species (ROS) homeostasis [17]. Oxidative stress is related to the pathogenesis of various diseases by damaging macromolecules such as membrane lipids, structural proteins, enzymes, and nucleic acids, leading to aberrant cell function and death [18]. In bacteria, selenoproteins have been frequently involved in energy conservation, among other functions [19]. Likewise, *Leishmania* parasites have also been reported to express several selenoproteins [20].

In this context, the role of Se in preventing cancer incidence and blocking tumor metastasis has been described [21]. Interestingly, Se was also found to play an important role in the pathophysiologic processes of certain clinical forms of leishmaniasis [22]. Further, Se and some selenocompounds have been proven to have a potential synergistic effect in combination with chemotherapeutic [23,24] or antileishmanial [25,26] drugs. Accordingly, several selenocompounds have been reported to encompass a broad spectrum of therapeutic activities, including anticancer [27,28], antibacterial [29,30], and antileishmanial properties [31,32]. Furthermore, we recently reported the synthesis of a series of functionalized acylselenoureas modulated by the inclusion of different small carbocyclic and heterocyclic scaffolds which were first evaluated for their antioxidant and anticancer properties [33]. Herein, *trans*-cinnamyl and benzo [*d*] [1,3] dioxolyl moieties were proven to be among the most appealing nuclei evaluated in this study. These scaffolds have also been reported for their anticancer [34,35,36], antileishmanial [37], and antioxidant [38] properties. As the aforementioned scaffolds exhibited favorable potential as antioxidant and anticancer agents, it was envisaged that the synthesis of new derivatives incorporating Se as other functionalities was worth the attempt, since the efficacy of selenocompounds as therapeutic drugs is highly correlated with the chemical form of this element [39]. The Se-based functional groups were selected considering previous studies in which these Se functionalities displayed a variety of biological activities. In this regard, selenocyanate, diselenide, and selenide derivatives have been well reported to possess anticancer [40,41], antileishmanial [42,43], and antibacterial [44,45] properties. However, the biological activity of these three Se forms in the same study has not been yet explored.

On the other hand, the inclusion of ferrocene was also considered. Ferrocene is a compact metallocene with low toxicity, high stability and reversible redox properties. Its incorporation among organic scaffolds has become a valuable toolkit for developing efficient and selective molecules to treat a variety of diseases [46], including cancer [47,48], leishmaniasis [49,50] and bacterial infection [51,52]. The combination of ferrocene and Se have also resulted in compounds with potential biological activities in recent years [53], although no study regarding antileishmanial activity has been reported to date to the best of our knowledge.

Thus, as an extension of our studies, in the current work, we focused on a synthetic approach based on the design of a novel series of cinnamyl and benzodioxyl derivatives comprising a variety of Se functionalities, with the aim to explore the combination of these Se forms with biological active scaffolds in the same framework. The antiproliferative, antileishmanial, antibacterial, and antioxidant properties of the new compounds were evaluated. To our knowledge, this is the first study that reports a comparative study including selenide-based compounds with ferrocene along with selenocyanate and diselenide analogs.

## 2. Results and Discussion

### 2.1. Chemistry and Characterization

The synthetic strategy for the preparation of the cinnamyl and benzodioxyl derivatives is outlined in Figure 1. The selenocyanate (SeCN) derivatives **1**–**2a** were obtained with high yields (>85% in both cases) by the nucleophilic substitution of the bromine (Br) atom by a SeCN group using KSeCN as the nucleophilic donor. The synthesis was achieved following a previously described procedure [54] in which the reaction was kept under a nitrogen atmosphere at room temperature with acetonitrile as solvent. The subsequent reduction of the selenocyanate compounds dissolved in ethanol with sodium borohydride at room temperature [55] yielded the diselenide derivatives **1**–**2b**. For ferrocene-containing selenides **1**–**2c**, (3-bromo-1-oxopropyl)ferrocene **5** was added to the mixture of the corresponding selenocyanate derivative with sodium borohydride prior to the workup of the reaction. The synthesis of intermediate **5** (Figure 1b) was carried out by a Friedel–Crafts acylation reaction on one of the cyclopentadienyl rings of the ferrocene moiety. The chloride derivative was synthesized initially, but no reaction was observed when it was added to the reaction with the selenocyanates **1**–**2a**, probably due to the less reactivity of the chlorine (Cl) atom. Additionally, previous attempts in obtaining the bromide derivative also failed to yield the monosubstituted ferrocene in a higher proportion than the disubstituted compound. This prompted us to seek an alternative route for the synthesis of (3-bromo-1-oxopropyl)ferrocene **5** by optimizing a previous method [56] with modifications. Thus, the synthesis of **5** was carried out by the reaction of ferrocene with 3-bromopropanoyl chloride in methylene chloride catalyzed by aluminum chloride (AlCl_3_) in 30 min. The product was purified by column chromatography with a gradient of hexane and ethyl acetate 95/5 and obtained in high yield (85%). To our knowledge, this is the first time that a rapid and simple method for the synthesis of selenides attached to a ferrocene scaffold is reported.

The structures of all the compounds were confirmed by ^1^H, ^13^C, and ^77^Se nuclear magnetic resonance (NMR) and elemental analysis, as described in the Experimental Section. The NMR spectra can be found in the Appendix A. Regarding ^1^H NMR, the signal belonging to the methylene group attached directly to the Se atom in both the cinnamyl and benzodioxyl derivatives appeared to be slightly shifted upfield as the chemical form of Se was modified from selenocyanate (3.91–4.25 ppm) to diselenide (3.70–3.84 ppm) and selenide (3.45–3.79 ppm), the peaks belonging to the benzodioxyl derivatives **2a**–**c** being in a higher range than the cinnamyl compounds with the same Se group. Likewise, the two methylene groups linking the ferrocene moiety to the Se atom in analogs **1**–**2c** were both shifted upfield to ~3 ppm when compared to the bromide intermediate **5** (~3.5 ppm) as a consequence of the substitution of the Br atom by a selenide group. This structural modification was also observed in the *J* values related to the coupling between both methylene groups, as it changed slightly from 6.6 Hz in **5** to 7.1 Hz in the cinnamyl and benzodioxyl derivatives **1**–**2c**. Concerning ^13^C NMR, the methylene group attached to the Se atom appeared in the same range (approximately 32 ppm) for derivatives **1**–**2a** and **1**–**2b**, regardless of the presence of the selenocyanate or diselenide groups, in contrast to the corresponding ^1^H NMR signals. Interestingly, a displacement of this signal was only observed in the case of the selenide compounds **1**–**2c**, in which the peaks appeared to be shifted upfield (~17 ppm). The peak belonging to the nitrile (CN) group in the selenocyanate compounds **1**–**2a** appeared in the range of 101.7–102.1 ppm, and its disappearance clearly indicated the formation of the diselenide or selenide derivatives, where this group was no longer present. Inspection of ^77^Se NMR spectra revealed a displacement in the position of the Se peak depending on the functional group in which the Se atom was incorporated in the molecule. Thus, in the case of the selenocyanate derivatives **1**–**2a**, this signal appeared at 264–284 ppm, whereas the peaks shifted downfield to a range of 364–401 ppm with the reduction to diselenide groups for compounds **1**–**2b**. Interestingly, the formation of the diselenide bond revealed the Se peaks most shifted downfield, while the analogs with a selenide group **1**–**2c** showed the opposite behavior, by having the lowest range among the series with Se peaks in the interval of 225–274 ppm. Overall, the benzodioxyl derivatives **2a**–**c** also tended to have the Se signals shifted downfield when compared to their cinnamyl counterparts **1a**–**c** in the three chemical forms of Se discussed in this work.

### 2.2. Biological Evaluation

#### 2.2.1. Antiproliferative Activity on Tumor and Nonmalignant Cells

The antiproliferative activities of the cinnamyl and benzodioxyl derivatives were assessed against six human cell lines of three different types of cancer (colon, lung, and breast) using the 3-(4,5-dimethylthiazol-2-yl)-2,5-diphenyltetrazolium bromide (MTT) assay at seven concentrations for 48 h. The results are expressed as the IC_50_ values and are outlined in Table 1. 

As shown in Table 1, the compounds displayed moderate antiproliferative activities, the exceptions being benzodioxyl derivatives **2a** and **2b**, with IC_50_ values below 12 µM in the colon (HT-29) and lung (H1299) cancer cell lines. Additionally, **2b** also exhibited cytotoxicity in the other colon cell line tested (HCT-116), with an IC_50_ value of approximately 11 µM. Interestingly, compounds **2a**–**b** were less active in the other lung cells (HTB-54) and in both the breast cell lines, **2b** being even completely inactive in MCF-7 cells. These results could imply a selectivity of these compounds towards certain cancer cell lines. On the other hand, cinnamyl derivative **1a** showed IC_50_ values above 50 µM in all the cell lines, whereas its diselenide analog **1b** showed moderate activity in the colon (HT-29 and HCT-116) and lung (H1299) cell lines with IC_50_ values below 40 µM. Surprisingly, the substitution of the selenocyanate or diselenide groups for a selenide attached to a ferrocene moiety led to a decrease in the activity in colon cells and a lung cell line (H1299), but derivatives **1**–**2c** proved to be more potent than their corresponding counterparts in the other lung cells (HTB-54), and in the breast cancer cells. In this context, the selenide derivatives **1**–**2c** displayed fair activity in MCF-7 cells with IC_50_ values below 20 µM, while the diselenides **1**–**2b** were inactive and the selenocyanates **1**–**2a** showed IC_50_ values above 50 µM. 

To assess their possible selectivity, the synthesized selenocompounds were then tested in a mammary gland (184B5) nonmalignant cell line, and the selectivity indexes (SI) were determined as the ratio of the IC_50_ values obtained for the nonmalignant cells and the homolog cancer cells. The results are reported in Table 2, along with the data for the breast cancer cells already shown in Table 1 for clarity.

Interestingly, compound **2b** was not only the most active compound with IC_50_ values below 12 µM in at least three cell lines, but it also did not display any activity in the nontumorigenic cells (Table 2). Additionally, the other diselenide **1b** showed the same behavior, as it was also not toxic in these cells. Derivative **2a** exhibited moderate activity with an IC_50_ value approximately 50 µM in the same cell line (184B5) and displayed a fair SI concerning the triple-negative breast cells (MDA-MB-231), this value being higher than the selenocyanate counterpart **1a**. On the contrary, selenides **1c** and **2c** were more selective when compared with the results regarding MCF-7 cells, with SI values of 1.3 and 3.4, respectively.

Given the small number of compounds tested, a detailed structure-activity relationship analysis is not feasible, but some observations and comparisons can be inferred. In this regard, taking into account the data for all the cell lines summarized in Table 1, the benzodioxyl moiety yielded in general more active compounds **2a**–**c** than their cinnamyl counterparts **1a**–**c**. In addition, this nucleus also yielded more selective compounds when comparing the nontumorigenic cell line (184B5) with the triple-negative breast cancer cells (MDA-MB-231) (Table 2). On the other hand, the presence of ferrocene in the structure did not improve significantly the cytotoxicity in comparison with the rest of the compounds of the series, but the combination with a selenide moiety yielded compounds that were more selective regarding the MCF-7 cells. Overall, the inclusion of Se in the chemical forms of selenocyanate, diselenide or selenide was determinant for the antiproliferative activity, yielding compounds with different cytotoxic profiles. 

#### 2.2.2. Antileishmanial Activity on Promastigote Forms

The cytotoxic activities of all the derivatives against *Leishmania major* and *Leishmania infantum* promastigotes were studied by the MTT assay after exposing the parasites to five different concentrations of each compound for 48 h. Miltefosine and paromomycin, two currently available drugs for the treatment of leishmaniasis, were used as positive controls. The IC_50_ values calculated from the dose–response curves are shown in Table 3. 

Collectively, compounds displayed a similar cytotoxic profile in both *Leishmania* forms, *L. infantum* promastigotes being more sensitive overall. Thus, selenocyanate-based compounds **1a** and **2a** proved to be the most active derivatives, with remarkable IC_50_ values below 5 µM in both cases. Moreover, the antileishmanial activity of **1**–**2a** was distinctly more potent than the cytotoxicity of the positive controls, especially in *L. major* since the IC_50_ values for the reference drugs were approximately 70 µM. Compounds **1**–**2b** containing a diselenide moiety exhibited moderate antileishmanial activity in both *Leishmania* forms, with higher IC_50_ values than the rest of the derivatives. However, although in *L. infantum* the IC_50_ values were higher compared to the IC_50_ values of the positive controls, in *L. major* both **1b** and **2b** were more active than miltefosine and paromomycin. Interestingly, our results showed that the selenocyanate derivatives **1**–**2a** were more active than their diselenide counterparts **1**–**2b** (Table 3). These data are contrary to other previous studies reported by our group [43,55], in which the diselenide moiety seemed to yield more active compounds. 

Additionally, the selenide-based compound **2c** showed potent antileishmanial activity in both promastigote forms, with IC_50_ values of 11.2 and 7.2 µM in *L. major* and *L. infantum*, respectively, these values being notably lower than those of the reference drugs miltefosine and paromomycin. To the best of our knowledge, this is the first study that reports a ferrocene-containing selenocompound with effective cytotoxic activity for the treatment of leishmaniasis. Further, its analog with a cinnamyl scaffold **1c** also displayed good antileishmanial activity with IC_50_ values approximately 25 µM in both promastigote forms, and lower when compared to those of the reference drugs in *L. major*. Accordingly, the cinnamyl derivatives yielded slightly less active compounds than the corresponding benzodioxyl derivatives overall, with the only exception of the diselenide **1b**. Therefore, these results suggest that the antileishmanial activity displayed by the new selenocompounds is not only the result of the presence of the Se group but also the combination with the organic scaffolds attached to it.

#### 2.2.3. Antibacterial Activity

The minimum inhibitory concentration (MIC) and the minimum bactericidal concentration (MBC) values of the selenocompounds were determined in a panel of six bacterial strains as a first approach to assess their antibacterial activity. Thus, the synthesized derivatives were tested against the Gram-negative bacteria *Escherichia coli* (*E. coli*), *Klebsiella pneumoniae* (*K. pneumoniae*) and *Citrobacter freundii* (*C. freundii*); and the Gram-positive bacteria *Staphylococcus aureus* (*S. aureus*), *Streptococcus faecalis* (*S. faecalis*), and *Staphylococcus epidermidis* (*S. epidermidis*). The compounds having MIC values higher than 200 µg/mL were considered inactive. The results are outlined in Table 4. 

As shown in Table 4, only the selenocyanate derivatives **1a** and **2a** displayed antibacterial activity against the six strains tested. Compound **1a** showed better inhibitory activity against Gram-positive bacteria, whereas the MIC and MBC values were 100 µg/mL against Gram-negative bacterial strains. Compound **2a** displayed also more potent antibacterial activity against Gram-positive bacteria, with MIC values equal to or lower than 25 µg/mL, although the MBC values could not be obtained for this compound against *S. aureus*, and no bactericidal effect was observed for both compounds against *S. faecalis*. This difference in the antibacterial activity displayed by **1**–**2a** between Gram-positive and Gram-negative bacterial strains could be related to the charge of the cell envelopes. Thus, Gram-negative bacteria would repeal the derivatives in a greater manner than the cell envelopes of Gram-positive cells, showing subsequently less inhibition of the bacterial growth in general. Compound **2a** was more active overall, with lower MIC values than its cinnamyl counterpart **1a**. In this context, **2a** was more active against four out of six bacterial strains (*E. coli*, *K. pneumoniae*, *S. aureus*, and *S. faecalis*). Compounds **1**–**2a** also exhibited remarkable antibacterial activity against *S. epidermidis*, with MIC and MBC values of 12.5 µg/mL. In contrast, **1**–**2a** were less active against *C. freundii*, with MIC and MBC values of 100 µg/mL. Interestingly, derivatives **1**–**2a** with a selenocyanate group were also the most active agents against *Leishmania* forms (Table 3). On the contrary, the diselenides (**1**–**2b**) and the ferrocene-containing selenides (**1**–**2c**) did not show any inhibitory effect even at the highest concentration evaluated (Table 4). Thus, the presence of the selenocyanate moiety seemed to be essential for the antibacterial activity of these compounds. These results are in concordance with other publications regarding the antibacterial activity of selenocyanate-containing compounds [57]. The difference in the activity between selenocyanate compounds (**1**–**2a**) and the other selenocompounds tested might be associated with the higher polarity of the selenocyanate derivatives, as their capacity to pierce the cell envelopes might be enhanced when compared with the less polar derivatives either with diselenide (**1**–**2b**) or selenide (**1**–**2c**) moieties. Nevertheless, the underlying mechanisms by which selenocyanates exert their antibacterial and antimicrobial effects remain unclear, although some proposals involving oxidative mechanisms have been recently published [58,59].

To further examine the inhibitory effect of the moderately active compounds **1a** and **2a** on the growth of the six bacterial strains tested, growth kinetic studies were performed. Thus, the six bacterial strains were exposed to three different concentrations of the test compounds using the MIC values (Table 4) as reference (2MIC, MIC, and MIC/2), while untreated cells were used as negative control. The bacterial growth was recorded by measuring the absorbance at several time points. The results are shown in Figure 1.

Overall, compounds **1a** and **2a** effectively inhibited the growth of the six bacterial strains evaluated. The complete inhibition could be observed at the highest concentration tested (2MIC) for both compounds, in which no bacterial growth was detected even after 48 h (Figure 1), thus suggesting a possible bactericidal effect. At lower concentrations, a delay in the bacterial growth of some strains could also be observed. In the case of *E. coli*, (Figure 1a,b) the delay was notable for both compounds at the lowest concentration evaluated (MIC/2). This delay was larger in the case of **1a** with a continuous lag phase of 41 h, while for compound **2a** this time was reduced to 10 h. Nevertheless, the concentrations of compound **2a** that were tested were lower than those of compound **1a**, given the results displayed in Table 4 in which compound **2a** was more active overall. The same behavior could also be observed in other bacterial strains. In this regard, bacterial growth was observed in *K. pneumoniae* (Figure 1c,d) when treated with 25 µg/mL and 50 µg/mL of compound **2a** with a log phase of 28 h and 42 h, respectively, while no growth was detected for compound **1a**. Likewise, in the case of the Gram-positive bacteria *S. aureus* (Figure 1g,h) and *S. faecalis* (Figure 1i,j), the exponential growth of bacteria was observed for the lowest concentrations tested of compound **2a** (12.5 µg/mL), these values (MIC/2) being lower than those of compound **1a** (25 µg/mL). In addition, the longest log phase was observed for compound **2a** at MIC concentration (25 µg/mL) in *S. aureus* (Figure 1g,h), with a delay in the bacterial growth of 44 h since the start of the experiments. Compound **2a** showed a log phase of 43 h at the lowest concentration (50 µg/mL) when compared with the data measured for compound **1a** in *C. freundii* (Figure 1e,f). On the other hand, as shown in Table 4, compounds were more active against the Gram-positive *S. epidermidis* strain. In this context, both compounds also clearly inhibited the bacterial growth at low concentrations in a time-dependent manner, while at the sub-MIC concentration tested (6.25 µg/mL) a delay could be observed (Figure 1k,l). Hence, compound **2a** led to the eventual bacterial growth in the six strains (Figure 1), despite having lower MBC values than those of compound **1a** (Table 4). This effect might be ascribed to the possible capacity of compound **2a** for inducing the formation of tolerant or persistent bacterial populations [60]. Nevertheless, more experiments would be needed to prove this hypothesis. 

Taken together, the results displayed in Table 4 and Figure 1 showed that the selenocyanate-containing compounds **1**–**2a** possess an inhibitory effect against several bacterial strains. Based on these data, further studies would be required to assess the potential use of selenocyanate-based compounds as antibacterial agents.

#### 2.2.4. Antioxidant Activity

Oxidative stress has been associated with a wide range of pathologies [18]. Thus, considering that the protection against oxidative injury is often suggested to be partly involved in the chemopreventive effects of Se [15], we evaluated the radical scavenging activity of the synthesized compounds using the 2,2-diphenyl-1-picrylhydrazyl (DPPH) assay as a preliminary study to test their potential antioxidant activity. Measurements were performed at 2 concentrations and recorded at eight different time points. Ascorbic acid (Asc) and Trolox were used as positive controls. The results are presented as the percentage of DPPH radical scavenging activity and are outlined in Figure 2.

The radical scavenging activity was time-dependent for all the compounds at the highest concentration tested (Figure 2a). On the contrary, activity was not completely dependent on the dose as expected initially, given that although an enhancement could be observed at 0.06 mg/mL compared to 0.03 mg/mL, this effect was not as relevant as time. Interestingly, only selenide compounds **1c** and **2c** displayed a potent antioxidant activity even at the lowest concentration tested (Figure 2b) and in a fast kinetic manner, reaching values almost comparable to those of the positive controls by the end of the assay (Figure 2a). Among them, benzodioxyl derivative **2c** was proved to be more active than **1c** at a low concentration and longer times (Figure 2b), although at a double concentration both compounds showed the same profile. The behavior of **1**–**2c** could probably be related to the presence of ferrocene attached to the selenide moiety, thus proving to be essential for the antioxidant activity of these compounds. In this context, selenocyanate (**1**–**2a**) and diselenide (**1**–**2b**) derivatives showed poor antioxidant activity with values below 10% even at 0.06 mg/mL (Figure 2a). Likewise, the cinnamyl derivatives **1a**–**b** seemed to yield slightly more active compounds than their benzodioxyl counterparts **2a**–**b** mainly at a high concentration (Figure 2a), differing from our previous results with these scaffolds [33] which included Se in an acylselenourea moiety. Overall, the antioxidant activity of the compounds tested in this work could be the result of the presence of ferrocene primarily, along with the combination of the chemical form of Se and the cinnamyl and benzodioxyl nuclei.

## 3. Materials and Methods

### 3.1. Chemistry

All chemicals were purchased from commercial suppliers and used as received without further purification throughout the course of experimental work. Reaction courses were monitored by thin-layer chromatography (TLC) developments on precoated silica gel 60 F254 aluminum sheets (Merck, Darmstadt, Germany), and the spots were visualized under UV light. The crude reaction products were purified by silica gel column chromatography using silica gel 60 Å (0.040–0.063 mm, Merck, Darmstadt, Germany) using hexane/ethyl acetate as the elution solvent. ^1^H, ^13^C, and ^77^Se NMR spectra were recorded on a Bruker Avance Neo 400 MHz operating at 400, 100, and 76 MHz, respectively, using TMS as the internal standard and CDCl_3_ as solvent. Chemical shifts are reported in δ values (ppm) and coupling constants (*J*) values are reported in hertz (Hz). Elemental analyses for carbon, hydrogen and nitrogen were performed on a Thermo Fisher FlashSmart™ Elemental Analyzer. Melting points (mp) were determined with a Mettler FP82 + FP80 apparatus (Greifensee, Switzerland). 

#### 3.1.1. General Synthesis of Selenocyanate Derivatives **1**–**2a**

Potassium selenocyanate (1.2 mmol) was added to a solution of the appropriate bromide derivative (1 mmol) in anhydrous acetonitrile (20 mL), and the reaction mixture was stirred at room temperature for 24 h under a nitrogen atmosphere. The resulting precipitate (KBr) was filtered off, and the filtrate was concentrated in vacuo to afford the desired product. 

*(E)-(3-selenocyanatoprop-1-en-1-yl)benzene* (**1a**). The title compound was synthesized from 3-bromo-1-phenyl-1-propene and potassium selenocyanate according to the general procedure described above. The compound was washed with hexane (2 × 20 mL) and a pink solid was obtained. Yield: 88%; mp: 87–89 °C. ^1^H NMR (400 MHz, CDCl_3_) δ 3.91 (d, 2H, *J* = 8.0 Hz, CH_2_); 6.37 (dt, 1H, *J* = 15.8 and 8.0 Hz, CH); 6.68 (d, 1H, *J* = 15.6 Hz, CH-Ph); 7.29 (d, 1H, *J* = 7.0 Hz, H_4_); 7.34 (t, 2H, *J* = 7.3 Hz, H_3_ + H_5_); 7.40 (d, 2H, *J* = 7.1 Hz, H_2_ + H_6_). ^13^C NMR (100 MHz, CDCl_3_) δ 32.1 (CH_2_), 101.7 (CN), 122.6 (CH-Ph), 126.9 (C_2_ + C_6_), 128.7 (C_4_), 128.8 (C_3_ + C_5_), 135.7 (CH), 136.2 (C_1_). ^77^Se NMR (76 MHz, CDCl_3_) δ 264. Analysis calcd for C_10_H_9_NSe (%): C, 54.06; H, 4.08; N, 6.30. Found: C, 53.95; H, 4.01; N, 6.22.

*5-(Selenocyanatomethyl)benzo[d]*[1,3]*dioxole* (**2a**). The title compound was synthesized from 5-(bromomethyl)-1,3-benzodioxole and potassium selenocyanate according to the general procedure described above. The compound was washed with hexane (2 x 20 mL) and a grey solid was obtained. Yield: 94%; mp: 71–73 °C. ^1^H NMR (400 MHz, CDCl_3_) δ 4.25 (s, 2H, CH_2_-Ph); 5.98 (s, 2H, O-CH_2_-O); 6.77 (m, 1H, H_3_); 6.83 (m, 2H, H_5_ + H_6_). ^13^C NMR (100 MHz, CDCl_3_) δ 33.4 (CH_2_-Ph), 101.6 (O-CH_2_-O), 102.1 (CN), 108.7 (C_3_), 109.2 (C_6_), 122.9 (C_5_), 129.0 (C_4_), 148.2 (C_2_), 148.3 (C_1_). ^77^Se NMR (76 MHz, CDCl_3_) δ 287. Analysis calcd for C_9_H_7_NO_2_Se (%): C, 45.02; H, 2.94; N, 5.83. Found: C, 45.28; H, 2.91; N, 5.75.

#### 3.1.2. General Synthesis of Diselenide Derivatives **1**–**2b**

The corresponding selenocyanate derivative **1a** or **2a** (1 mmol) was dissolved in absolute ethanol (25 mL), and sodium borohydride (1.5 mmol) was added to the solution. The reaction was stirred for 1 h at room temperature. The solvent was then removed under vacuum by rotatory evaporation, and the crude product was treated with water and extracted with methylene chloride (3 × 20 mL). The organic layer was washed with water (3 × 30 mL) and dried over anhydrous sodium sulfate to afford the final product. 

*1,2-Dicinnamyldiselenide* (**1b**). The title compound was synthesized from compound **1a** and sodium borohydride according to the general procedure described above. The product was purified by column chromatography over silica gel using a gradient of hexane and ethyl acetate to obtain a white solid. Yield: 21%; mp: 78–80 °C. ^1^H NMR (400 MHz, CDCl_3_) δ 3.70 (d, 4H, *J* = 8.0 Hz, CH_2_, CH_2′_); 6.22 (dt, 2H, *J* = 15.7 and 8.0 Hz, CH, CH’); 6.39 (d, 2H, *J* = 15.6 Hz, CH-Ph, CH’-Ph); 7.24 (m, 2H, H_4_, H_4′_); 7.27 (m, 8H, H_2_ + H_3_ + H_5_ + H_6_, H_2′_ + H_3′_ + H_5′_ + H_6′_). ^13^C NMR (100 MHz, CDCl_3_) δ 32.4 (CH_2_, CH_2′_), 126.0 (CH-Ph, CH’-Ph), 126.5 (C_2_ + C_6_, C_2′_ + C_6′_), 127.7 (C_4_, C_4′_), 128.8 (C_3_ + C_5_, C_3′_ + C_5′_), 132.6 (CH, CH’), 136.9 (C_1_, C_1′_). ^77^Se NMR (76 MHz, CDCl_3_) δ 364. Analysis calcd for C_18_H_18_Se_2_ (%): C, 55.11; H, 4.63. Found: C, 55.31; H, 4.59.

*1,2-Bis(benzo[d]*[1,3]*dioxol-5-ylmethyl)diselenide* (**2b**). The title compound was synthesized from compound **2a** and sodium borohydride according to the general procedure described above. The compound was washed with hexane (2 × 20 mL) and a yellow solid was obtained. Yield: 31%; mp: 78–80 °C. ^1^H NMR (400 MHz, CDCl_3_) δ 3.84 (s, 4H, CH_2_-Ph, CH_2′_-Ph); 5.95 (s, 4H, O-CH_2_-O, O-CH_2′_-O); 6.70 (dd, 2H, *J* = 7.9 and 1.6 Hz, H_5_, H_5′_); 6.74 (m, 4H, H_3_ + H_6_, H_3′_ + H_6′_). ^13^C NMR (100 MHz, CDCl_3_) δ 33.0 (CH_2_-Ph, CH_2′_-Ph), 101.2 (O-CH_2_-O, O-CH_2′_-O), 108.3 (C_3_, C_3′_), 109.5 (C_6_, C_6′_), 122.4 (C_5_, C_5′_), 132.9 (C_4_, C_4′_), 146.9 (C_2_, C_2′_), 147.8 (C_1_, C_1′_). ^77^Se NMR (76 MHz, CDCl_3_) δ 401. Analysis calcd for C_16_H_14_O_4_Se_2_ (%): C, 44.88; H, 3.30. Found: C, 45.07; H, 3.29.

#### 3.1.3. Synthesis of (3-bromo-1-oxopropyl)ferrocene (**5**)

The intermediate was prepared following a previously published procedure [56] with modifications. Briefly, 3-bromopropanoyl chloride (0.54 mL, 1 mmol) was added to a suspension of ferrocene (1 g, 1 mmol) in methylene chloride (25 mL). Aluminum chloride (0.80 g, 1.1 mmol) was then added to the mixture, and the resulting dark purple reaction was allowed to stir for 30 min at room temperature. The homogeneous solution was firstly washed with water (2 × 20 mL) and then with a saturated aqueous solution of potassium bicarbonate (3 × 20 mL). The organic layer was dried over anhydrous sodium sulfate and concentrated in vacuo. The crude product of the reaction was purified using silica gel chromatography with a gradient of hexane and ethyl acetate. Unreacted ferrocene was firstly eluted at 100% hexane, followed by product **5** at 95/5 ratio; finally, if any, the disubstituted derivative could be eluted separately. An orange solid was obtained. Yield: 85%, mp: 63–65 °C. ^1^H NMR (400 MHz, CDCl_3_) δ 3.31 (t, 2H, *J* = 6.6 Hz, CH_2_-C=O); 3.74 (t, 2H, *J* = 6.6 Hz, CH_2_-Br); 4.26 (s, 5H, Cp_fc_); 4.54 (t, 2H, *J* = 1.9 Hz, H_fc_ + H_fc_); 4.80 (t, 2H, *J* = 2.0 Hz, H_fc_ + H_fc_). ^13^C NMR (100 MHz, CDCl_3_) δ 26.0 (CH_2_-Br), 42.3 (CH_2_-C=O), 69.3 (2C_fc_), 69.9 (5Cp_fc_), 72.6 (2C_fc_), 78.3 (C_fc_-C=O), 200.7 (C=O).

#### 3.1.4. General Synthesis of Ferrocene-Containing Selenide Derivatives **1**–**2c**

Sodium borohydride (1.5 mmol) was added to a suspension of the corresponding selenocyanate **1a** or **2a** (1 mmol) in absolute ethanol (25 mL). The (3-bromo-1-oxopropyl)ferrocene **5** (1 mmol) was added in small portions with caution to the solution when the selenocyanate was completely dissolved, and the reaction mixture was stirred for 2-3 h at room temperature. The product was then extracted with methylene chloride (3 × 20 mL) and washed with water (2 × 30 mL). The organic layers were dried over anhydrous sodium sulfate and concentrated in vacuo. 

*3-(Cinnamylselanyl)-1-ferrocenylpropan-1-one* (**1c**). The title compound was synthesized from compound **1a,** sodium borohydride and (3-bromo-1-oxopropyl)ferrocene according to the general procedure described above. The product was purified by column chromatography over silica gel using a gradient of hexane and ethyl acetate to obtain an orange solid. Yield: 82%; mp: 64–66 °C. ^1^H NMR (400 MHz, CDCl_3_) δ 2.88 (t, 2H, *J* = 7.1 Hz, CH_2_-Se); 3.12 (t, 2H, *J* = 7.1 Hz, CH_2_-C=O); 3.45 (d, 2H, *J* = 7.5 Hz, CH_2_); 4.20 (s, 5H, Cp_fc_); 4.48 (m, 2H, H_fc_ + H_fc_); 4.76 (m, 2H, H_fc_ + H_fc_); 6.34 (dt, 1H, *J* = 15.5 and 7.6 Hz, CH); 6.45 (d, 1H, *J* = 15.7 Hz, CH-Ph); 7.22 (t, 1H, *J* = 7.3 Hz, H_4_); 7.31 (t, 2H, *J* = 7.5 Hz, H_3_ + H_5_); 7.39 (d, 2H, *J* = 7.4 Hz, H_2_ + H_6_). ^13^C NMR (100 MHz, CDCl_3_) δ 17.1 (CH_2_), 26.6 (CH_2_-Se), 41.0 (CH_2_-C=O), 69.4 (2C_fc_), 70.0 (5Cp_fc_), 72.5 (2C_fc_), 78.7 (C_fc_-C=O), 126.4 (C_2_ + C_6_), 127.0 (CH-Ph), 127.6 (C_4_), 128.7 (C_3_ + C_5_), 131.5 (CH), 136.9 (C_1_), 202.7 (C=O). ^77^Se NMR (76 MHz, CDCl_3_) δ 225. Analysis calcd for C_22_H_22_FeOSe (%): C, 60.43; H, 5.07. Found: C, 60.22; H, 5.08.

*3-((Benzo[d]*[1,3]*dioxol-5-ylmethyl)selanyl)-1-ferrocenylpropan-1-one* (**2c**). The title compound was synthesized from compound **2a,** sodium borohydride and (3-bromo-1-oxopropyl)ferrocene according to the general procedure described above. The product was purified by column chromatography over silica gel using a gradient of hexane and ethyl acetate to obtain a red solid. Yield: 20%; mp: 76–79 °C. ^1^H NMR (400 MHz, CDCl_3_) δ 2.83 (t, 2H, *J* = 7.1 Hz, CH_2_-Se); 3.03 (t, 2H, *J* = 7.2 Hz, CH_2_-C=O); 3.79 (s, 2H, CH_2_-Ph); 4.20 (s, 5H, Cp_fc_); 4.49 (m, 2H, H_fc_ + H_fc_); 4.74 (m, 2H, H_fc_ + H_fc_); 5.92 (s, 2H, O-CH_2_-O); 6.72 (d, 1H, *J* = 7.9 Hz, H_6_); 6.77 (dd, 1H, *J* = 7.9 and 1.6 Hz, H_5_); 6.86 (d, 1H, *J* = 1.6 Hz, H_3_). ^13^C NMR (100 MHz, CDCl_3_) δ 17.4 (CH_2_-Ph), 28.0 (CH_2_-Se), 40.7 (CH_2_-C=O), 69.3 (2C_fc_), 69.9 (5Cp_fc_), 72.4 (2C_fc_), 78.6 (C_fc_-C=O), 101.1 (O-CH_2_-O), 108.2 (C_3_), 109.4 (C_6_), 122.0 (C_5_), 133.3 (C_4_), 146.5 (C_2_), 147.9 (C_1_), 202.7 (C=O). ^77^Se NMR (76 MHz, CDCl_3_) δ 274. Analysis calcd for C_21_H_20_FeO_3_Se (%): C, 55.41; H, 4.43. Found: C, 55.59; H, 4.33.

### 3.2. Biology

#### 3.2.1. Cell Culture Conditions

The cell lines were obtained from the American Type Culture Collection (ATCC). All the cancer cell lines (HT-29, HCT-116, H1299, HTB-54, MDA-MB-231 and MCF-7) and the nontumorigenic cells (184B5) were maintained in RPMI 1640 medium (Gibco), supplemented with 10% fetal bovine serum (FBS; Gibco) and 1% antibiotics (10.00 units/mL penicillin and 10.00 mg/mL streptomycin; Gibco). Cells were preserved in tissue culture flasks at 37 °C and 5% CO_2_. Culture medium was replaced every three days.

#### 3.2.2. Cell Viability Assay

The cytotoxic effect of each compound was initially tested at seven different concentrations ranging between 0.5 and 100 µM in the cell lines mentioned above. The cells were seeded (1 × 10^4^ cells per well) in 96-well plates and incubated at 37 °C and 5% CO_2_ for 24 h. After attachment, cells were treated with either dimethyl sulfoxide (DMSO) or increasing concentrations of the corresponding selenocompound for 48 h. The effect of the compounds on cell viability was determined by the MTT assay [33]. In brief, 20 µL of MTT (5 mg/mL) was added to each well 2.5 h prior to experiment termination. At the end of the incubation, the medium was removed, and the formazan crystals were dissolved in 50 µL of DMSO. Absorbance was measured at 550 nm on a microplate reader. IC_50_ values were calculated using OriginPro 8.5.1. software by nonlinear curve fitting. Selectivity indexes were calculated as the ratio of the IC_50_ values determined for the nonmalignant and the tumoral cells in the breast cell lines (IC_50_ (184B5)/IC_50_ (MDA-MB-231) and IC_50_ (184B5)/IC_50_ (MCF-7)). Data were obtained from at least three independent experiments performed in triplicates.

#### 3.2.3. Culture Conditions of Leishmania Promastigotes

*L. major* (clone VI, MHOM/IL/80/Friendlin) and *L. infantum* (BCN-150) promastigotes were grown in continuous stirred M199 1X medium (Sigma, St Louis, MO, USA) at 26 °C supplemented with 25 mM HEPES (pH 7.2), 10% heat-inactivated FBS (Gibco, Gaithersburg, MD, USA), 0.1 mM adenine, 0.0005% (w/v) hemin, 0.0001% (w/v) biotin, 100 UI/mL penicillin and 100 mg/mL streptomycin (Gibco, Gaithersburg, MD, USA). Media were changed before each experiment to achieve exponential growth.

#### 3.2.4. Parasite Viability Assay

The antileishmanial activity was assessed by the MTT assay as previously described [61]. Briefly, 3 × 10^6^ promastigotes of both *Leishmania* strains were placed into a 96-well plate with supplemented M199 1X medium during their logarithmic growth phase. The parasites were treated with five increasing concentrations of the compounds ranging between 0.1 and 1000 µM at 26 °C for 48 h. At the termination point, 20 µL of MTT (5 mg/mL) was added and further incubated for 4 h. The resultant formazan crystals formed were dissolved in 80 µL of DMSO and the absorbance was measured at 550 nm on a microplate reader. IC_50_ values were calculated using OriginPro 8.5.1. software by nonlinear curve fitting. Miltefosine and paromomycin were used as reference drugs. Data were obtained from at least three independent experiments performed in triplicates.

#### 3.2.5. Antibacterial Activity: MIC and MBC Assays

All the selenocompounds were screened for their in vitro inhibitory activity against three Gram-negative (*E. coli*, *K. pneumoniae* and *C. freundii*) and three Gram-positive (*S. aureus*, *S. faecalis* and *S. epidermidis*) bacteria. Bacterial strains were kindly provided by the Department of Microbiology and Parasitology of the University of Navarra. The bacterial cells were grown in tryptic soy agar (TSA) plates and maintained at 4 °C. 

The compounds were firstly dissolved in DMSO and serially diluted in a 96-well plate in tryptic soy broth (TSB) with eleven concentrations ranging from 400 µg/mL to 0.4 µg/mL with a final volume of 100 µL. Subsequently, 100 µL of a diluted suspension of bacterial cells with an optical density of 0.125 measured at 600 nm in TSB (inoculum of approximately 10^5^ CFU/mL) was added to each well and incubated at 37 °C for 24 h. After the incubation period, each well was analyzed for the presence or absence of visual growth of bacterial cells. The lowest concentration of the test compound at which no visible growth occurred was determined as the MIC values. After incubation, 10 µL of the bacterial suspension from each well that presented antibacterial activity was added to divided TSA plates and incubated at 37 °C overnight. The concentrations that did not show any microorganism growth were considered as the MBC values. 

#### 3.2.6. Growth Curve Studies

A stock solution of selected compounds **1a** and **2a** was prepared in DMSO, and different concentrations equivalent to 2MIC, MIC, and MIC/2 in TSB of each compound were dispensed into a 96-well plate with a final volume of 100 µL, while only TSB was added for the control wells. Then, 100 µL of a diluted suspension of bacterial cells in TSB with an optical density of 0.125 measured at 600 nm (inoculum of approximately 10^5^ CFU/mL) was added to each well, and the incubation was monitored by spectrophotometric measurements. The absorbance was recorded at 600 nm against time (h) at predetermined periods on a microplate reader. The experiments were performed in triplicate.

#### 3.2.7. DPPH Radical Scavenging Assay

The antioxidant activity of the selenocompounds was determined by the DPPH assay following a procedure previously published by our group [33]. Briefly, the compounds were dissolved in absolute methanol at a concentration of 1 mg/mL, and a dilution was prepared. Asc and Trolox were used as positive controls. A methanolic solution (100 µM) of DPPH˙ was made daily and protected from the dark. The stability of the radical through the time of analysis was checked by absorbance measurements of this solution. A volume of 100 µL of DPPH in methanol was added to 100 µL of the compounds’ solutions to a final concentration of 0.03 or 0.06 mg/mL, and the decolorization of the purple radical to the yellowish reduced form was followed by recording the absorbance at 517 nm. Determinations were recorded at eight different time intervals from 0 to 120 min on a BioTeck PowerWave XS spectrophotometer and the data were collected using KCJunior software. All the measurements were carried out in triplicate. Results are expressed as the percentage of the radical scavenged, calculated using the following formula:$$\%\;\mathrm{DPPH}\;\mathrm{radical}\;\mathrm{scavenging}=\frac{A_{\mathrm{control}}{\;–\;A}_{\mathrm{sample}}}{A_{\mathrm{control}}}\;\times \;100$$
where A_control_ refers to the absorbance of the negative control and A_sample_ refers to the absorbance of the tested compounds.

#### 3.2.8. Statistical Analysis

Data were expressed as the mean ± SD (standard deviation) and experiments were performed at least thrice in triplicates unless otherwise specified. Nonlinear curve regression analysis calculated by OriginPro 8.5.1. software was used to assess the IC_50_ values. The two-way analysis of variance (ANOVA) was used to calculate the statistical significance of differences by comparing the enhancement in the percentage of DPPH radical scavenging activity at different time points for the same compound. Data were analyzed using GraphPad Prism version 8.0.1., and the statistically significant values (*p*-value) for ANOVA analysis were taken as **** *p* < 0.0001, *** *p* < 0.001, ** *p* < 0.01, and * *p* < 0.05 when comparing consecutive time points for compound **1c**; and ^####^
*p* < 0.0001, ^###^
*p* < 0.001, ^##^
*p* < 0.01, and ^#^
*p* < 0.05 when comparing consecutive time points for compound **2c**.

## 4. Conclusions

In summary, a series of cinnamyl and benzodioxyl derivatives bearing Se in the chemical forms of selenocyanate, diselenide, or selenide were synthesized and evaluated in vitro as antiproliferative, antileishmanial, and antibacterial agents, along with their potential antioxidant activity. The inclusion of a ferrocene moiety in the structure was also considered. Benzodioxyl derivatives **2a**–**b** showed fair antiproliferative activity with IC_50_ values below 12 µM in colon (HT-29) and lung (H1299) cancer cell lines, while being selective towards nonmalignant breast cells. Likewise, among the selenocompounds included in this work, selenocyanate derivatives **1**–**2a** displayed potent antileishmanial activity against promastigote forms of *L. major* and *L. infantum*, with notably IC_50_ values below 5 µM in both cases, and much lower than those of reference drugs (71.0 μM > IC_50_ > 18.0 μM). Selenocyanate derivatives **1–2a** also displayed moderate antibacterial activity against several Gram-positive and Gram-negative bacterial strains and had an inhibitory effect on the growth of the six bacterial strains tested, with particularly low MIC and MBC values (12.5 µg/mL) in *S. epidermidis*. Additionally, the ferrocene-containing selenide **2c** was identified as a potent antileishmanial agent with IC_50_ values of 11.2 and 7.2 µM in *L. major* and *L. infantum*, respectively, this being the first selenocompound with a ferrocene moiety to be reported as a leishmanicidal derivative in the literature to date. The ferrocene-containing selenide derivatives **1**–**2c** also displayed potent radical scavenging activity with values almost comparable to those of the positive controls after two hours at 0.06 mg/mL (72.6% and 76.1%, respectively), while the rest of compounds showed no significant effect. Overall, the benzodioxyl derivative **2a** including a selenocyanate moiety was delineated as a multitarget compound with antiproliferative, antileishmanial, and antibacterial properties. Thus, the results generated in this study could encourage the synthesis of new Se derivatives based on the structural features of compound **2a** along with their biological evaluation to obtain agents with an appealing therapeutic profile. 

## Data Availability

All data presented in this study are available within this article and Appendix A.

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
