# Peer review of "Unveiling a New Selenocyanate as a Multitarget Candidate with Anticancer, Antileishmanial and Antibacterial Potential"

_molecules, 2022, doi:10.3390/molecules27217477_

Round 1
Reviewer 1 Report
The paper "Unveiling a new selenocyanate as a multitarget candidate with 2 anticancer, antileishmanial and antibacterial potential" is good and logical. And also, it is significant to the biological activity. This paper has been reviewed but it needs minor revision before accepted. The followings are the points need to modify.
1. The abbreviation needs to be shown in the first place where it is showed.
2. Fig. 1 should be enhanced
3. The unit of all the paper should be in uniform.
4. The conclusion needs to be more specific results of the paper.
Author Response
We attach the response to reviewer 1

Reviewer 2 Report
The article "Unveiling a new selenocyanate as a multitarget candidate with anticancer, antileishmanial and antibacterial potential" reports the synthesis and anticancer, antileishmanial, antibacterial and antioxidant properties of a series of organoselenic compounds, including seleno-ferrocenic compounds. The article is written in a very clear manner, from the introduction, synthesis and characterization of the compounds, the antiproliferative evaluations against six human cell lines of three different types of cancer (colon, lung, and breast), the antileishmanial, antibacterial and antioxidant evaluations, up to conclusions.
The article deserves to be published in its present form.

Author Response
We attach the file with the response to reviewer 2

Reviewer 3 Report
Abstract:
There is no background/introduction in the abstract. Please add a line/statement of introduction/background.
It would be better if the authors mention the name/type of chemistry reaction behind the synthesis in the abstract.
Line 1, Authors have written “Se”. When writing for the first time, please write the complete name i.e. “Selenium (Se)”.
Introduction:
The introduction is well written overall. I would prefer to cut a few sentences to slightly shorten it to hold the reader’s interest.
References: There are 73 references in the manuscript which are a bit high. I would urge the authors to reduce the number by deleting unnecessary ones especially in the introduction.
Overall, the manuscript titled “Unveiling a new selenocyanate as a multitarget candidate with anticancer, antileishmanial and antibacterial potential” is a good piece of work. The authors have synthesized new compounds that have exhibited multitarget efficacies. The manuscript is well written. The data supports the conclusion. I would recommend it for publication after minor revisions.
Author Response
We attach the file with the response to reviewer 3.
